# Safety and Efficacy of Intravenous Alteplase before Endovascular Thrombectomy: A Pooled Analysis with Focus on the Elderly

**DOI:** 10.3390/jcm11133681

**Published:** 2022-06-26

**Authors:** Asaf Honig, Hen Hallevi, Naaem Simaan, Tzvika Sacagiu, Estelle Seyman, Andrei Filioglo, Moshe J. Gomori, Ofer Rotschild, Tali Jonas-Kimchi, Udi Sadeh, Anat Horev, Ronen R. Leker, José E. Cohen, Jeremy Molad

**Affiliations:** 1Department of Neurology, Hadassah-Hebrew University Medical Center, Jerusalem 91120, Israel; tzvika.sacagiu@yahoo.com (T.S.); afilioglo@gmail.com (A.F.); leker@hadassah.org.il (R.R.L.); 2Department of Stroke and Neurology, Tel Aviv Sourasky Medical Center, Tel Aviv 6423906, Israel; henh@tlvmc.gov.il (H.H.); seyman.estelle@gmail.com (E.S.); ofer.rotschild@gmail.com (O.R.); jeremymolad@gmail.com (J.M.); 3Department of Neurology, Ziv Medical Center, Safed 13100, Israel; naaems@ziv.gov.il; 4Department of Medical Imaging, Hadassah-Hebrew University Medical Center, Jerusalem 91120, Israel; gomori@cc.huji.ac.il; 5Department of Medical Imaging, Tel Aviv Sourasky Medical Center, Tel Aviv 6423906, Israel; talijk@tlvmc.gov.il (T.J.-K.); udisad@tlvmc.gov.il (U.S.); 6Department of Neurology, Soroka University Medical Center, Beer Sheva 84417, Israel; anat.horev8@gmail.com; 7Department of Neurosurgery, Hadassah-Hebrew University Medical Center, Jerusalem 91120, Israel; jcohenns@yahoo.com

**Keywords:** acute ischemic stroke, elderly, endovascular thrombectomy, intravenous alteplase

## Abstract

Current guidelines advocate intravenous thrombolysis (IVT) prior to endovascular thrombectomy (EVT) for all patients with acute ischemic stroke (AIS) due to large vessel occlusion (LVO). We evaluated outcomes with and without IVT pretreatment. Our institutional protocols allow AIS patients presenting early (<4 h from onset or last seen normal) who have an Alberta Stroke Program Early CT Score (ASPECTS) ≥6 to undergo EVT without IVT pretreatment if the endovascular team is in the hospital (direct EVT). Rates of recanalization and hemorrhagic transformation (HT) and neurological outcomes were retrospectively compared in consecutive patients undergoing IVT+EVT vs. direct EVT with subanalyses in those ≥80 years and ≥85 years. In the overall cohort (IVT+EVT = 147, direct EVT = 162), and in subsets of patients ≥80 years (IVT+EVT = 51, direct EVT = 50) and ≥85 years (IVT+EVT = 19, direct EVT = 32), the IVT+EVT cohort and the direct EVT group had similar baseline characteristics, underwent EVT after a comparable interval from symptom onset, and reached similar rates of target vessel recanalization. No differences were observed in the HT frequency, or in disability at discharge or after 90 days. Patients receiving direct EVT underwent more stenting of the carotid artery due to stenosis during the EVT procedure (22% vs. 6%, *p* = 0.001). Direct EVT and IVT+EVT had comparable neurological outcomes in the overall cohort and in the subgroups of patients ≥80 and ≥85 years, suggesting that direct EVT should be considered in patients with an elevated risk for HT.

## 1. Introduction

Endovascular thrombectomy (EVT) is considered the recommended mode of reperfusion therapy for large vessel occlusion (LVO) in the anterior circulation in patients with acute ischemic stroke (AIS) [1]. As intravenous thrombolysis (IVT) was administered before EVT (IVT+EVT) in the large EVT clinical trials [2], current guidelines [1] advocate IVT treatment prior to EVT in all cases of AIS due to LVO. Moreover, a meta-analysis of 13 observational studies and a recent large observational study [3] showed that when compared to EVT alone, patients who received IVT+EVT had better functional outcomes, lower mortality, and a higher rate of successful recanalization requiring fewer device passes, as well as equal odds of symptomatic intracerebral hemorrhage (sICH) [4].

Two recent randomized clinical trials, the Endovascular Thrombectomy With or Without Intravenous Alteplase in Acute Stroke (DIRECT-MT) [5] and the Effect of Endovascular Treatment Alone vs Intravenous Alteplase Plus Endovascular Treatment on Functional Independence in Patients With Acute Ischemic Stroke (DEVT) trials [6]—both from China—showed noninferiority of direct EVT as compared with IVT+EVT. However, another clinical trial, MR CLEAN-NO IV [7], failed to show either superiority or noninferiority of the direct EVT approach.

Given the modest efficacy of IVT for large vessel occlusions, our primary goal was to examine its contribution to both reperfusion targets and patient functional outcome. In addition, as the relevant clinical trials [5,6,7,8] included a negligible number of patients older than 80 years, we aimed to compare direct EVT with IVT+EVT in an elderly subpopulation.

## 2. Materials and Methods

### 2.1. Study Population

Our two academic centers’ institutional protocols allow patients presenting in the early stage of AIS (<4 h from onset or last seen normal) who have an Alberta Stroke Program Early CT Score (ASPECTS) [9] ≥6 to undergo direct EVT without preceding IVT if the interventional team is in the hospital. Perfusion imaging is not mandatory. We compared clinical and procedural outcomes, safety, and workflow between patients treated with both IVT and EVT (IVT+EVT group) and those treated with EVT alone (direct EVT group) in routine clinical practice.

Consecutive AIS patients who underwent EVT for LVO in the two participating academic centers are included in the centers’ prospective ongoing databases. For this study, patients who underwent EVT for internal carotid artery (ICA), M1 middle cerebral artery (MCA), tandem ICA-MCA, basilar, anterior cerebral artery (ACA), proximal M2 MCA, or P1 posterior cerebral artery (PCA) occlusions were included. Patients with occlusions distal to the M2 segment of the MCA or P1 segment of the PCA were excluded from the study. Also excluded were patients who were treated with anticoagulation and therefore not eligible to receive IVT (Appendix A). Additionally, patients who showed major improvement with resolution of their main deficits following IVT and were not taken to digital subtraction angiography (DSA) if repeat CTA showed resolution of the previously seen vessel occlusion were excluded from the study. We performed a separate analysis of intention-to-treat with the inclusion of patients who showed vessel recanalization following IVT only. Data from the two centers for the patients treated between 1 January 2016 and 30 December 2020 who met the inclusion criteria were pooled for retrospective analysis for this study. The study was approved by the institutional review boards of the participating centers with a waiver of informed consent.

### 2.2. EVT Treatment Algorithm

All included patients underwent EVT using stentrievers and aspiration techniques or combined approaches at the discretion of the treating endovascular specialist. All patients were treated with similar institutional protocols, including intensive care unit admissions post-procedure. All had repeat noncontrast head CT (NCCT) 24 h post-procedure. If indicated, those who underwent thrombectomy alone began antiplatelet or anticoagulation treatment after NCCT. Some patients in the direct EVT group also underwent a carotid stent implant for the treatment of stenosis during the EVT procedure at the discretion of the treating physicians. Stenting was performed after administration of a loading dose of aspirin (300 mg) and clopidogrel (300 mg). Clopidogrel and aspirin reactivities were measured at the time of endovascular intervention with the VerifyNow method [10]. Clopidogrel reactivity was defined as P2Y12 reaction units (PRU) <150 [11]. Patients who underwent stent placement from the IVT+EVT group received a loading dose of aspirin only when the clopidogrel loading dose was added following the 24 h follow-up NCCT that ruled out HT.

### 2.3. Data Collection

We collected demographic and vascular risk factors. Neurological deficits were measured using the National Institutes of Health Stroke Scale (NIHSS) [12] at admission and discharge. Stroke etiology was classified with the TOAST classification [13]. Time metrics and imaging variables, including ASPECTS before treatment, were assessed. Collateral status was assessed on admission CTA according to the ASPECT collateral grading scale [14], with a score of 4–5 defined as good collaterals. Data on procedural variables, including the modified Thrombolysis in Cerebral Infarction (mTICI) score [15] at the end of the procedure and the number of passes needed to achieve the best possible recanalization, were also studied. mTICI2b-3 was considered a successful target vessel recanalization.

Hemorrhagic transformation (HT) was assessed both radiologically and clinically according to the ECASS-2 criteria [16]. We used post-EVT NCCT data to classify HT into petechial hemorrhagic infarction (HI) and parenchymal hematoma (PH) type 1 or 2, defined as confluent hemorrhage covering less or more than 1/3 of the infarct volume, respectively [17]. A dual-energy CT protocol was used to enable a distinction between HT and contrast extravasation due to blood–brain barrier damage. Imaging studies were reviewed by experienced stroke neurologists and neuroradiologists blinded to the clinical scenario, and the degree of HT was adjudicated by discussion until a consensus was reached. Further clinical division was made into asymptomatic and symptomatic ICH (sICH). sICH was defined as any apparent extravascular blood in the brain or within the cranium that was associated with clinical deterioration (defined as an increase of 4 points or more in the score on the NIHSS), or led to death, and was identified as the predominant cause of the neurologic deterioration [18].

Following the acute neurocritical care period, all patients underwent a thorough investigation to assess the underlying etiology for the stroke, thus enabling a tailored stroke prevention treatment. The investigation for all patients included a 24–48 h EKG Holter and transthoracic echocardiogram. All patients were examined by a stroke neurologist on follow-up at the stroke prevention clinic. Most patients underwent MRI including MRA after their discharge. In stroke patients ≤65, an echocardiogram was performed with a test of agitated saline to detect a patent foramen ovale. Both a transesophageal echocardiogram and a prolonged EKG Holter were performed at the discretion of the stroke neurologist. Treatment with anticoagulation was initiated whenever a cardioembolic etiology was identified.

Functional outcome was assessed with the modified Rankin Score (mRS) [19] prior to stroke, upon discharge, and 90 days after stroke. A good functional outcome was defined as either an mRS ≤ 2 for patients with baseline mRS ≤ 2, or mRS = 3 in patients that had mRS = 3 upon admission. Ninety-day mortality was used as a secondary outcome parameter.

### 2.4. Statistical Analysis

Statistical analysis was performed using SPSS 27 (IBM, Armonk, NY, USA), and *p* < 0.05 was considered significant. The chi-square test was used to explore the link between qualitative variables. Student’s *t*-test was used to compare quantitative variables. The Mann–Whitney test was used to compare differences between two independent groups when the dependent variable was either ordinal or continuous, but not normally distributed. Included in the multivariate models were age, sex, and predictors that were found to be significant in the univariate analysis. The first analysis included only patients that underwent the EVT procedure. A second per-protocol analysis included patients with the initial intention to treat based on the vessel occlusion in the initial imaging—namely, patients who were initially intended to be treated with IVT+EVT. However, these patients showed vessel recanalization post-IVT, making EVT unnecessary.

## 3. Results

Overall, 309 patients (age 70.72 ± 14.7, 34.6% males) were included in the study (Table 1). Patients from Hadassah (n = 183) and Tel Aviv Ichilov (n = 129) medical centers were similar in age (71 ± 15 vs. 71 ± 14, *p* = 0.9) and sex (male 46% vs. 47%, *p* = 0.9). The IVT+EVT cohort (*p* = 147) and the direct EVT group (n = 162) had similar baseline characteristics, including age (mean 71 vs. 70, respectively), sex, vascular risk factors, admission NIHSS (15.6 vs. 16.2), baseline mRS (0.8 vs. 0.79), and underlying etiology (59% cardioembolic etiology for both groups) (Table 1).

Overall, direct EVT and IVT+EVT patients had comparable times from symptom onset or last seen normal to emergency department (ED) presentation (86.7 ± 62.2 vs. 90.3 ± 60.5 min, *p* = 0.6), and from symptom onset to groin puncture (237 vs. 247.6 min, *p* = 0.6). ED door to groin puncture time was similar in the IVT+EVT group (159 vs. 149 min, *p* = 0.552), while imaging to groin puncture time tended to be longer in the IVT+EVT group (137 vs. 119 min, *p* = 0.326).

Direct EVT patients underwent more carotid stenting during the EVT procedure (*p* = 0.001) and had better post-procedural improvement in NIHSS (*p* = 0.039), but no difference was observed in NIHSS or mRS at discharge or the 90-day follow-up. There was no difference in the rates of HT overall or the HT subtypes.

A total of 101 patients ≥80 years of age (mean 85.8 ± 4.5 years, 27% males) were included in the study (Table 2). Compared with patients <80 years of age, they had a higher baseline mRS (1 [0–3] vs. 0 [0], *p* < 0.001), admission NIHSS (19 [14–22] vs. 15 [10–20], *p* < 0.001), and mRS 90 (2 [1–4] vs. 4 [3–6], *p* < 0.001), and lower rates of favorable outcome (28% vs. 53%, *p* < 0.001) (Appendix A). Patients above 80 years of age also had higher rates of hypertension and atrial fibrillation and lower rates of smoking (82% vs. 61%, *p* < 0.001, 49% vs. 31%, *p* = 0.03, 13% vs. 34%, *p* < 0.001, respectively). Notably, the rates of bridging IVT administered in patients ≥80 years and those <80 years were comparable (50.5% vs. 46.1%, *p* = 0.474).

In patients ≥80 years, baseline characteristics and stroke severity were similar for the direct EVT and IVT-EVT groups, apart from higher rates of previous antiplatelet treatment (46% vs. 22%, *p* = 0.016) in the direct EVT group (Table 2). Carotid stenting was used more frequently in the direct EVT group (12% vs. 2%, *p* = 0.04). Rates of favorable outcome (33% vs. 27%, *p* = 0.558) and mortality (22% vs. 22%, *p* = 0.958) were similar for the two groups (Table 3). Surprisingly, the direct EVT group had a higher rate of ICH PH2 (6% vs. 0%, *p* = 0.045), but there were comparable rates of sICH (4.3% vs. 0%, *p* = 0.16).

A total of 51 patients ≥85 years of age were included in the study. In this age group, patients who underwent direct EVT and those who had IVT+EVT had comparable rates of TICI2b-3 (68% vs. 83%, *p* = 0.243), good functional outcome (37% vs. 28%, *p* = 0.519), mortality (54% vs. 46%, *p* = 0.418), PH2 (4.3% vs. 0%, *p* = 0.384), and sICH (0% vs. 0%).

A total of 208 patients < 80 years of age (mean 63.3 ± 12.2 years, 56% males) were included in the study (Appendix A) with similar baseline characteristics and prognostic outcomes between the IVT+EVT and direct EVT groups.

A multivariate analysis for predictors of good functional outcome was performed for the entire cohort. Age (OR 1.034 [1.01–1.06] per year decrease), baseline NIHSS (OR 1.063 [1.01–1.11] per unit decrease), and baseline mRS (OR 1.51 [1.17–1.95] per unit decrease) were found to be significant predictors.

### Per-Protocol Analysis

Recanalization was achieved after IVT administration in nine patients, and EVT was aborted in four of them before arterial puncture due to significant clinical improvement (average NIHSS drop from 9 to 2) and clot resolution on repeat CTA. Five other patients underwent diagnostic angiography, which demonstrated recanalization (TICI 2b in two patients with distal M3/4 emboli and TICI 3 in three patients). All nine patients suffered from MCA occlusion (five M1 segment, four M2 segment). A new analysis including these patients was performed (n = 318), with no significant changes in the analysis of outcome measures, including 90-day good functional outcome, mRS, HT, and mortality.

## 4. Discussion

Our pooled analysis comparing patients who had similar baseline characteristics, stroke etiology, severity, and treatment timeframes, and who were treated with direct EVT or IVT-EVT, found similar rates of recanalization and prognostic outcomes across all age groups. Our findings from real-world data corroborate the findings of recent large clinical trials from China, the DEVT [6] and DIRECT-MT [5] trials, which showed noninferiority of the direct EVT approach. Our findings are in contrast with the MR CLEAN-NO IV [7] large clinical trial, which failed to show such noninferiority. While in our study, rates of achieving TICI 2b−3 were similar for the two groups, comparable recanalization rates were lower for the direct EVT group in the MR CLEAN-NO IV [7] trial based on assessment at both final procedure DSA and CTA performed 24 h post-procedure. The advantages in outcomes for IVT+EVT in these studies were quite possibly driven by lower rates of recanalization and ultimately reperfusion.

Surprisingly, in our per-protocol analysis, IVT alone achieved early recanalization in only nine patients (5.7%). This finding demonstrates the low yield of IVT as a stand-alone therapy in LVO and points to a need for re-evaluation of the common “drip and ship” algorithm for LVO patients in centers not equipped for endovascular therapy rather than direct transfer to thrombectomy-capable centers. Advantages of IVT bridging may include higher EVT reperfusion rates and decrease rates of procedure-related remote emboli [20]. However, the similar rates of successful reperfusion, first-pass recanalization, and number of passes for recanalization found in our cohort suggest that IVT had no significant impact on the EVT procedure in our patients. Furthermore, there was no indication that IVT reduced the frequency of distal emboli, since the rates of TICI 2b3, as well as NIHSS and mRS at discharge and the 90-day follow-up, were comparable.

IVT utility and safety greatly vary between various stroke etiologies, and a patient-tailored approach for the use of bridging IVT has thus been suggested [21]. In our study, subanalyses for patients aged ≥80 and ≥85 years who had similar baseline and stroke characteristics found comparable rates of recanalization and good functional outcome scores for the two treatment strategies. This suggests that age should not serve as a main factor in the decision for or against bridging therapy. Future studies should be directed towards further assessment of this issue, as well as identifying the patients who would benefit most from direct EVT vs. IVT+EVT.

The main concern with IVT is the rate of HT. Previous studies describe higher rates of HT post-IVT among older patients [22]. This may be attributed to a higher burden of white matter hyperintensities and cerebral microbleeds with increasing age [23]. In our observational study, alteplase treatment was not associated with higher rates of HT in the overall population or, importantly, in the elderly subpopulation. Similar findings were reported in the DIRECT-MT study, which found that baseline NIHSS and admission glucose levels were independent risk factors for SCH after EVT, while IVT did not appear to increase the risk [24].

It is noteworthy that in the DIRECT-MT trial, a subset of patients who were pretreated with alteplase and then required more than three passes to achieve recanalization during EVT had an independent association with sICH when compared with patients managed with direct EVT. This finding may direct us towards a subset of patients who are at higher risk of developing a sICH. Bridging with IVT may also increase the risk of sICH in the presence of early ischemic changes [25]. This may be related to longer procedure times, resulting in a higher burden of ischemic tissue. Patients with delayed ED arrival and early CT ischemic changes may be more prone to developing sICH, and managing these patients with the IVT–EVT approach should be carefully considered.

Importantly, in our study, rates of HT and its subtypes were not increased among older patients who were treated with IVT+EVT vs. direct EVT. Moreover, among patients ≥80 years, direct EVT was associated with higher rates of PH2, and a nonsignificant trend for sICH [24]. As previously suggested [26], we postulate that the most powerful predictors of sICH are the extent of ischemic changes and the degree of LVO recanalization, which are directly connected with recanalization rates rather than age.

The conflicting results of previous trials discussing bridging IVT may point to the need for a new patient-tailored approach to determining when IVT should be administered. The results of this study suggest that age should not be a key factor in patient selection. As some trials found a beneficial effect for IVT prior to EVT, and as our study shows a comparable risk for ICH, IVT should be considered mainly when the EVT team is not immediately available and in patients without elevated risk for major bleeding. However, in some scenarios, direct EVT is preferable. For example, in both participating centers, more patients in the direct EVT group underwent carotid stenting. This may point to the benefit of a direct EVT approach among patients with severe symptomatic carotid stenosis or tandem lesions, since more patients from the direct EVT group were able to undergo carotid stent placement with the required immediate post-procedure double antiplatelet therapy. Additionally, in patients with a relative contraindication to alteplase, a readily available direct EVT approach may be reasonable. Further studies are needed to better determine a patient selection algorithm for bridging therapy in terms of both efficacy and safety.

Our study has several limitations. It was an observational study and not a randomized clinical trial. In addition, selection bias cannot be ruled out as patients were selected for IVT according to the discretion of the attending neurologist specializing in stroke management. Finally, this study did not include more distal vessel occlusions that may show higher rates of recanalization in response to IVT and therefore benefit more from an IVT–EVT approach. This study’s strengths include a real-world scenario that encompassed all EVT-eligible patients and thus included a large elderly population.

## Figures and Tables

**Table 1 jcm-11-03681-t001:** Characteristics of patients in the direct EVT and IVT+EVT groups.

	Direct EVT n = 162	IVT+EVT n = 147	*p* Value
Age (SD)	70.1 (16.0)	71.4 (13.5)	0.419
Sex, male (%)	71 (43.8)	37 (25.2)	0.305
Hypertension (%)	112 (69.1)	96 (65.3)	0.474
Diabetes (%)	47 (29)	47 (32)	0.572
Hyperlipidemia (%)	73 (45.1)	69 (46.9)	0.741
Smoking (%)	39 (24.1)	46 (31.3)	0.156
Atrial fibrillation (%)	62 (38.3)	55 (37.4)	0.877
Ischemic heart disease (%)	47 (29)	55 (37.4)	0.117
Valve disease (%)	7 (4.3)	11 (7.5)	0.236
Congestive heart failure (%)	11 (6.8)	10 (6.8)	0.658
Chronic renal failure (%)	10 (6.2)	9 (6.1)	0.963
Prior stroke (%)	21 (13)	22 (15)	0.611
Malignancy (%)	22 (13.6)	10 (6.8)	0.051
Statins (%)	43 (26.5)	30 (20.4)	0.47
Antiplatelets (%)	51 (31.5)	36 (24.5)	0.198
**Occlusion vessel (%)**			0.189
ICA	45 (27.8)	34 (23.1)	
M1 MCA	87 (53.7)	79 (53.7)	
M2 MCA	18 (11.1)	24 (16.3)	
Basilar	6 (3.7)	8 (5.4)	
ACA	3 (1.9)	1 (0.7)	
PCA	0 (0)	0 (0)	
Tandem lesion	29 (17.9)	18 (12.4)	0.052
Side, right (%)	83 (55.3)	63 (55.7)	0.946
**TOAST (%)**			0.968
Cardioembolism	96 (59.3)	87 (59.2)	
Large-artery atherosclerosis	36 (22.2)	28 (19)	
Other determined etiology	8 (4.9)	3 (2)	
Undetermined	21 (13)	25 (17)	
Collaterals (IQR)	3 (1–4)	3 (2–4)	0.211
**Time frames (minutes)**			
Symptom to groin puncture (SD)	237.0 (201.4)	247.6 (141.6)	0.6
Symptom to door (SD)	86.7 (62.2)	90.3 (60.5)	0.608
Door to imaging (SD)	34.0 (51.4)	23.4 (23.7)	**0.022**
Door to groin puncture (SD)	148.9 (184.8)	158.6 (127.1)	0.552
Imaging to groin puncture (SD)	118.8 (178.3)	136.6 (122.2)	0.326
**EVT procedure characteristics**			
Number of passes (IQR)	1 (1–3)	1 (1–3)	0.698
TICI 2b-3 (%)	131 (83.4)	121 (84)	0.89
First pass	71 (49)	49 (50.5)	0.813
Stent	32 (21.8)	8 (6.1)	**<0.001**
**Complications and outcomes**			
HT	28 (18.4)	26 (18.2)	0.856
HT PH2	8 (6.1)	6 (4.2)	0.486
Symptomatic ICH	6 (4.5)	2 (1.6)	0.236
NIHSS on admission (IQR)	16 (11–21)	16 (11–20)	0.469
NIHSS on discharge (IQR)	14 (2–10)	5 (1–9)	0.886
mRS baseline (IQR)	0 (0–2)	0 (0–2)	0.622
mRS discharge (IQR)	3 (2–5)	4 (2–5)	0.13
mRS 90 (IQR)	3 (1–5)	3 (2–5)	0.449
Mortality (%)	20 (12.3)	21 (14.3)	0.616
Outcome (favorable)	76 (46.9)	58 (39.4)	0.103

*Values represent number of patients unless otherwise stated. Bold numbers signify statistically significant p values. SD = standard deviation; IQR = interquartile range; ICA = internal carotid artery; MCA = middle cerebral artery; ACA = anterior cerebral artery; PCA = posterior cerebral artery; HT = hemorrhagic transformation; PH = parenchymal hematoma; mRS = modified Rankin Scale*.

**Table 2 jcm-11-03681-t002:** Characteristics of patients aged ≥80 years in the direct EVT and IVT+EVT groups.

	Direct EVT N = 50	IVT+EVT N = 51	*p* Value
Age (SD)	86.9 (5.1)	84.7 (3.9)	0.106
Sex, male (%)	11 (22)	16 (31.4)	0.287
Hypertension (%)	43 (86)	38 (74.5)	0.147
Diabetes (%)	16 (32)	15 (29.4)	0.778
Hyperlipidemia (%)	21 (42)	26 (51)	0.366
Smoking (%)	5 (10)	8 (15.7)	0.394
Atrial fibrillation (%)	28 (56)	22 (43.1)	0.196
Ischemic heart disease (%)	17 (34)	20 (39.2)	0.586
Valvular disease (%)	1 (2)	2 (3.9)	0.57
Congestive heart failure (%)	6 (12)	4 (7.8)	0.312
Chronic renal failure (%)	7 (14)	2 (3.9)	0.065
Prior stroke (%)	8 (16)	5 (9.8)	0.353
Malignancy (%)	6 (12)	1 (2)	**0.047**
Statins (%)	17 (34)	13 (25.5)	0.765
Antiplatelets (%)	23 (46)	11 (21.6)	**0.016**
**Occlusion vessel (%)**			
ICA	15 (30)	12 (23.5)	0.23
M1 MCA	28 (56)	28 (54.9)	
M2 MCA	5 (10)	10 (19.6)	
Basilar	1 (2)	1 (2)	
ACA	0 (0)	0 (0)	
PCA	0 (0)	0 (0)	
Tandem lesion	9 (18)	5 (9.8)	0.128
Side, right (%)	28 (56)	24 (47.1)	0.677
**TOAST (%)**			
Cardioembolism	35 (70)	33 (64.7)	0.57
Large-artery atherosclerosis	8 (16)	8 (15.7)	0.96
Other determined etiology	1 (2)	0	
Undetermined	5 (10)	9 (17.6)	
Collaterals (IQR)	2 (1–4)	3 (1–4)	0.771
Symptom to door (SD)	100.9 (67.9)	89.8 (56.4)	0.11
Door to imaging (SD)	34.3 (49.4)	22.5 (30.2)	0.06
Symptom to groin puncture (SD)	227.8 (148.1)	221.5 (85.7)	**0.028**
Door to groin puncture (SD)	123.1 (131.0)	134.3 (59.2)	0.079
**EVT procedure characteristics**			
Number of passes (IQR)	1 (1–3)	1 (1–3)	0.839
TICI 2b-3 (%)	37 (74)	40 (78.4)	0.566
First pass			
Stent	6 (12)	1 (2)	**0.04**
**Complications and outcomes**			
HT	10 (20)	5 (9.8)	0.158
HT PH2	3 (6)	0 (0)	**0.045**
Symptomatic ICH	2 (4)	0 (0)	0.162
NIHSS on admission (IQR)	18 (14–22)	19 (14–22)	0.921
NIHSS on discharge (IQR)	7 (2–14)	5 (1–14)	0.43
mRS baseline (IQR)	1 (0–3)	2 (0–3)	0.592
mRS discharge (IQR)	4 (3–5)	4 (3–5)	0.924
mRS 90 (IQR)	4 (3–6)	4 (3–6)	0.869
Mortality (%)	11 (22)	11 (21.6)	0.958
Outcome (favorable)	15 (30)	13 (25.5)	0.558

*Values represent number of patients unless otherwise stated. Bold numbers signify statistically significant p values. SD = standard deviation; IQR = interquartile range; ICA = internal carotid artery; MCA = middle cerebral artery; ACA = anterior cerebral artery; PCA = posterior cerebral artery; HT = hemorrhagic transformation; PH = parenchymal hematoma; mRS = modified Rankin Scale*.

**Table 3 jcm-11-03681-t003:** Comparison of outcomes in patients aged <80 years and ≥80 years, in the direct EVT and IVT+EVT groups.

	Symptomatic HT			Discharge NIHSS (IQR)			mRS 90 Days (IQR)			Good Functional Outcome			Death		
Patient Age	Direct EVT	IVT+EVT	*p*	DirectEVT	IVT+EVT	*p*	DirectEVT	IVT+EVT	*p*	DirectEVT	IVT+EVT	*p*	Direct EVT	IVT+EVT	*p*
**<80 years**	4	2	0.59	3	5	0.78	2	3	0.35	61	45	0.15	9	10	0.55
**(n = 208)**	(3.7%)	(2.4%)		(1–9)	(2–8)		(1–4)	(2–4)		(58%)	(48%)		(8%)	(10%)	
**≥80 years**	2	0	0.16	6	5	0.43	4	4	0.87	15	13	0.56	11	11	0.96
**(n = 101)**	(4.3%)	(0%)		(2–14)	(1–14)		(3–6)	(3–6)		(33%)	(27%)		(22%)	(22%)	

*Values represent number of patients unless otherwise stated. IQR = interquartile range*.

## Data Availability

The data presented in this study are available on request from the corresponding author. The data are not publicly available due to the requirements of the institutional review boards.

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
