# Peer review of "Safety and Efficacy of Intravenous Alteplase before Endovascular Thrombectomy: A Pooled Analysis with Focus on the Elderly"

_jcm, 2022, doi:10.3390/jcm11133681_

Round 1

Reviewer 1 Report

The work is a retrospective comparative analysis of the effectiveness of protocols for treating patients with direct endovascular thrombectomy (EVT) or with the addition of bridged intravenous thrombolysis (IVT). In total, medical histories, tests, treatment protocols and examinations of 309 patients were studied, including elderly ones (over 80 and over 85 years old). As a result, there were no statistically significant differences in treatment outcomes and risks in different study groups. This leads to the need to consider modifying the “drip and ship” protocol, at least for some patients that have, for example, additional risks connected with SICH. Perhaps in some cases it is acceptable and even more justified to use a faster direct EVT protocol in order to reduce the procedure time, which will possibly save more patients. The authors of the manuscript have done serious work, performed a multivariate retrospective analysis using statistical methods. In my opinion, the work deserves evaluation by the medical expert community. The expert has no serious comments, and minor flaws can be corrected as part of editorial analysis and proofreading.

Author Response

We thank reviewer 1 for his comments. We have made sure a revision of the manuscript was done by a scientific editor.

Reviewer 2 Report

This study evaluated recanalization rates and neurological outcomes between IVT+EVT and direct EVT. The authors showed that direct-EVT and IVT+EVT had comparable neurological outcomes in the overall-cohort and in subgroups of patients ≥80 and ≥85 years. This study suggests that direct-EVT should be considered in patients with elevated risk for HT. Here are my suggestions:

1.      Please describe Materials and Methods using subheadings.

2.      In occlusion distal to M2 or P1, IVT may be superior to direct DVT. Patients who have resolved with IVT alone or have not undergone DSA may be favorable to IVT. However, since this study excluded these patients, a bias in favor of direct EVT may exist.

3.      The method of registering IVT+EVT patients is not clearly described. Were a similar number of IVT+EVT patients enrolled in this study from both centers? Were the study subjects enrolled consecutively, randomly, or matching? What actions have been taken to reduce selection bias?

4.      Was there no difference in stroke evaluation depth between the two centers of enrolled patients?

5.      The definitions of inclusion and exclusion are vague. Resolve these issues by presenting a patient flow chart as a figure.

6.      After the procedure, according to what guidelines were stroke prevention treatment performed? Please describe clearly.

7.      Is the anticoagulant treatment similar between IVT+EVT and direct EVT?

8.      In Table 1, describe the unit of time frames. Please add abbreviations to the table's footnotes.

9.      What are the clinical characteristics between ≥80 and <80 years and between IVT+EVT and EVT at <80 years? Please present these data as supplemental tables.

10.   The methodology mentioned that multivariate analysis was performed, but the results of multivariate analysis are absent. In addition, the authors used a very ambiguous intention to treat analysis. If it is ITT, did you estimate the sample size considering statistical power when selecting the number of two groups?

Author Response

Reviewer 2

 This study evaluated recanalization rates and neurological outcomes between IVT+EVT and direct EVT. The authors showed that direct-EVT and IVT+EVT had comparable neurological outcomes in the overall-cohort and in subgroups of patients ≥80 and ≥85 years. This study suggests that direct-EVT should be considered in patients with elevated risk for HT. Here are my suggestions:

Please find all changes we have made according to the reviewer's comments highlighted in yellow throuout  

  1. Please describe Materials and Methods using subheadings.

Done

  1. In occlusion distal to M2 or P1, IVT may be superior to direct DVT. Patients who have resolved with IVT alone or have not undergone DSA may be favorable to IVT. However, since this study excluded these patients, a bias in favor of direct EVT may exist.

We are grateful for the comment. Our study does show the benefit of the direct thrombectomy approach only in the LVO patients. We have therefore added the following comment to limitations of the study in the discussion section: Finally, the study does not include more distal vessel occlusions that may show higher rates of recanalization in response to IVT and therefore benefit more from IVT-EVT approach.

  1. The method of registering IVT+EVT patients is not clearly described. Were a similar number of IVT+EVT patients enrolled in this study from both centers? Were the study subjects enrolled consecutively, randomly, or matching? What actions have been taken to reduce selection bias?

We thank the reviewer for the important methodological comment.

Both centers enrolled consecutive large vessel occlusion patients who were eligible to receive intravenous thrombolysis prior to their endovascular thrombectomy procedure. By enrolling consecutive patients and by having similar baseline and stroke characteristics between the Direct-EVT and the IVT-EVT groups we minimize possible bias. As mentioned int the methods section:" Consecutive AIS patients who underwent EVT for LVO in the two participating academic centers are included in the centers’ prospective ongoing databases."

Additionally, as mentioned in the methods section, both medical centers keep a similar institutional treatment algorithm. Moreover, patients collected from both centers had similar age, sex, ratio of patients aged ≥80, and similar baseline mRS. We have therefore added the following description the results section: Patients from Hadassah (n=183) and Tel-Aviv-Ichilov (n=129) medical centers were similar in age (71±15 vs 71±14, p=0.9) and sex (male 46% vs 47%, p=0.9).

  1. Was there no difference in stroke evaluation depth between the two centers of enrolled patients?

Stroke evaluation in terms of degree and etiology is not different between the two centers. The tools to assess clinical and radiological presentation and outcome are outlined under "Data Collection" subheading of the methods section.

  1. The definitions of inclusion and exclusion are vague. Resolve these issues by presenting a patient flow chart as a figure.

We are grateful for pointing it out for us. We have now added a flow chart to the supplemental material.

  1. After the procedure, according to what guidelines were stroke prevention treatment performed? Please describe clearly.

We have added the following paragraph to the methods section:

Following the acute neurocritical care period, all patients underwent a thorough investigation to assess the underlying etiology for the stroke, thus enabling a tailored stroke prevention treatment. The investigation for all patients included a 24-48-hour EKG Holter and trans-thoracic-echocardiogram. All patients were examined by a stroke neurologist on follow-up at the stroke prevention clinic. Most patients underwent MRI including MRA after their discharge. In stroke patients ≤65 echocardiogram was performed with a test of agitated saline to detect Patent-Foramen-Ovale. Both trans-esophageal-echocardiogram and a prolonged EKG Holter were performed at the discretion of the stroke neurologist. Treatment with anticoagulation was initiated whenever a cardioembolic etiology was identified.

  1. Is the anticoagulant treatment similar between IVT+EVT and direct EVT?

In order to avoid possible bias patients who were treated with anticoagulation upon their admission and therefore contraindicated to receive IVT were excluded from the study.

To clarify we have added the following statement to the methods section under the subheading of Study Population as well as in the patient selection flowchart in the supplemental material: Also excluded were patients who were treated with anticoagulation and therefore not eligible to receive IVT (Graph 1, supplemental material).

  1. In Table 1, describe the unit of time frames. Please add abbreviations to the table's footnotes.

We have made the requested changes.

  1. What are the clinical characteristics between ≥80 and <80 years and between IVT+EVT and EVT at <80 years? Please present these data as supplemental tables.

We thank the reviewer for this important comment. The two tables have been added to the supplemental material as requested.

  1. The methodology mentioned that multivariate analysis was performed, but the results of multivariate analysis are absent.

We apologize for our mistake as the results have been accidently omitted. A multivariate model was built using variables such as age, sex and all variables associated with good functional outcome in univariate model. We have therefore added the following sentences to the results section:

In a multi-variant analysis for predictors of good functional outcome was performed for the entire cohort. Age (OR 1.034 [1.01-1.06] per year decrease), baseline NIHSS (OR 1.063 [1.01-1.11] per unit decrease) and baseline mRS (OR 1.51[1.17-1.95] per unit decrease) were found to be significant predictors.

  1. In addition, the authors used a very ambiguous intention to treat analysis. If it is ITT, did you estimate the sample size considering statistical power when selecting the number of two groups?

Again, we apologize for not accurately defining the analysis. Instead of ITT we have now renamed it as per protocol analysis. Namely, patients who were initially intended to be treated with both IVT and EVT, however, due to vessel recanalization due to IVT, EVT was not deemed necessary. We added to the methods section: A second per-protocol analysis included patients with the initial intention to treat based on the vessel occlusion in the initial imaging. Namely, patients who were initially intended to be treated with IVT+EVT, however, showed vessel recanalization post IVT, making EVT unnecessary.

Round 2

Reviewer 2 Report

Please present the results of multivariable analysis as a table or supplemental material.

Author Response

The supplementary is attached
